# Nonlinear dynamics and chaos in an optomechanical beam

Daniel Navarro-Urrios[1], Néstor E. Capuj[2], Martín F. Colombano[1], P. David García[1], Marianna Sledzinska[1], Francesc Alzina[1], Amadeu Griol[3], Alejandro Martínez[3] & Clivia M. Sotomayor-Torres[1,4]

Optical nonlinearities, such as thermo-optic mechanisms and free-carrier dispersion, are often considered unwelcome effects in silicon-based resonators and, more specifically, optomechanical cavities, since they affect, for instance, the relative detuning between an optical resonance and the excitation laser. Here, we exploit these nonlinearities and their intercoupling with the mechanical degrees of freedom of a silicon optomechanical nanobeam to unveil a rich set of fundamentally different complex dynamics. By smoothly changing the parameters of the excitation laser we demonstrate accurate control to activate two- and four-dimensional limit cycles, a period-doubling route and a six-dimensional chaos. In addition, by scanning the laser parameters in opposite senses we demonstrate bistability and hysteresis between two- and four-dimensional limit cycles, between different coherent mechanical states and between four-dimensional limit cycles and chaos. Our findings open new routes towards exploiting silicon-based optomechanical photonic crystals as a versatile building block to be used in neurocomputational networks and for chaos-based applications.

[1] Catalan Institute of Nanoscience and Nanotechnology (ICN2), CSIC and The Barcelona Institute of Science and Technology, Campus de la Universidad Autónoma de Barcelona, Edifici ICN2, Bellaterra, Barcelona 08193, Spain. [2] Depto. Física, Universidad de la Laguna, La Laguna 38206, Spain. [3] Nanophotonics Technology Center, Universitat Politècnica de València, Valencia 46022, Spain. [4] Catalan Institute for Research and Advances Studies ICREA, Barcelona 08010, Spain. Correspondence and requests for materials should be addressed to D.N.-U. (email: daniel.navarro@icn2.cat or danielnavarrourrios@gmail.com).

The long-term solutions of any one- or two-dimensional nonlinear system restrict to nothing more complicated than a stable/unstable fixed point or a limit cycle, their possible bifurcation types being already sufficiently well studied[1]. If more than two dimensions are at play, trajectories may become much more complex, eventually displaying aperiodicity and extreme sensibility to initial conditions, that is, exhibiting chaos. The Lorenz equations are the paradigmatic example of how complex and rich the solution of a relatively simple deterministic set of differential equations can be[2,3], where only one of the three is nonlinear. Despite being five decades old, wide ranges of the parameters governing the equations are still unexplored.

Classical nonlinear dynamics have been studied quite extensively in different optomechanical (OM) architectures, where the classical version of quantum Langevin equations describes the evolution of the optical cavity field and the mechanical oscillator[4]. A relatively standard solution of that nonlinear system is a state of wide mechanical coherent oscillations[5]. Interesting features and tools can appear in that regime, for example, the existence of several stable mechanical-amplitude solutions for a fixed set of external parameters, which was studied theoretically in ref. 6 and demonstrated experimentally in refs 7,8. Moreover, as predicted in ref. 9 for sufficiently high laser power, the system can eventually enter into a chaotic regime, which may be exploited for chaos-based secure data communication or sensing, among other applications. In this regard, integrated OM systems may present advantages with respect to coupled lasers[10] or hybrid optoelectric oscillators[11] in terms of the ease of integration, scalability and engineering of the nonlinearities[12]. Despite their potential, only the works of Carmon et al.[13,14] and Monifi et al.[15] in dielectric microtoroids and by Wu et al.[16] and our group[17] in silicon (Si)-based integrated devices have addressed chaos in integrated OM architectures.

We report here on the nonlinear dynamics of an OM cavity system, namely a Si-based one-dimensional photonic crystal, mainly affected by optical nonlinearities of diverse origin: thermo-optic effects, free-carrier dispersion and OM coupling[18]. These are intercoupled through the number of intracavity photons ($n$), which affects and is affected by the previous mechanisms. As a consequence, new dynamics in an OM cavity are disclosed by exploring a wide area of the parameter space. The system displays a heterogeneous variety of stable dynamical solutions that, in some specific cases, coexist and give rise to bistability and hysteresis.

## Results

### General properties of the sample. The device investigated here is a one-dimensional OM photonic crystal fabricated using standard Si nanofabrication processes (see Methods) in a silicon-on-insulator (SOI) wafer. As seen from the top view in Fig. 1a, the crystal lattice constant is quadratically reduced towards the centre of the beam, hereby defining high-$Q$ optical cavity modes.

To model accurately the fabricated OM system and account for the differences with respect to the nominal geometry, the in-plane geometry is imported from the scanning electron microscopy (SEM) micrographs into the finite element method (FEM) solver (see Methods and Supplementary Fig. 4), the thickness being that of the top Si layer of the SOI wafer. This procedure ensured a good agreement between the measured optical modes and those extracted from simulations (Fig. 1b).

In the following, we investigate the third optical mode (Fig. 1c) of the OM photonic crystal. It displays an asymmetric field distribution with respect to the $xz$ plane, giving rise to surprisingly high values of the single-particle OM coupling rate ($g_{o,OM}$, dots of Fig. 1d) for in-plane ($xy$ plane) flexural modes (red

dots of Fig. 1d). In Fig. 1d, we have indicated with an arrow the one having three antinodes along the $x$-direction ($\Omega_m = 54$ MHz), since it will be shown it is at the heart of the complex dynamics discussed in the next sections. The radio-frequency (RF) signal associated with optical transduction of thermally driven mechanical eigenmodes is also reported in Fig. 1d. The peaks positions show a good agreement with the modes predicted to display high $g_{o,OM}$ values.

An evident discrepancy between the FEM simulations of $g_{o,OM}$ and the actual transduced RF signal concerns out-of-plane flexural modes (blue dots of Fig. 1d), which should not present significant $g_{o,OM}$ values due to symmetry considerations (see Methods and Supplementary Fig. 6). However, geometric inhomogeneities along the $z$ axis may break the symmetry along that direction, unbalancing the contributions from the top and bottom Si–air interfaces. As it will be discussed later, among the out-of-plane flexural modes, the fundamental one ($\Omega'_m = 5$ MHz, first peak in the RF spectrum) plays a key role in the chaotic dynamics of the system.

**Bistability and hysteresis.** It is well known that large energies stored in optical resonators may give rise to strong nonlinear effects. The two most intense sources of optical nonlinearities in Si-based optical resonators are: thermo-optic effects, which redshift the resonance position ($\lambda_r$) from its unperturbed position ($\lambda_o$) proportionally to the average cavity temperature increase ($\Delta T$); and free-carrier dispersion, which induces a blueshift proportional to the free carrier population ($N$). The dynamics of $\Delta T$ and $N$ can be described by a system of rate equations (see Methods) that are coupled through the number of intracavity photons:

$$n = n_o \frac{\Delta\lambda_o^2}{4(\lambda_l - \lambda_r)^2 + \Delta\lambda_o^2}, \tag{1}$$

where $n = n_o = 2P_l\kappa_e\lambda_o/\kappa^2 hc$ in perfect resonance, that is, $\lambda_l = \lambda_r$. $P_l$ and $\lambda_l$ are the laser power and wavelength, respectively; $\kappa_e$ and $\kappa$ are the extrinsic and overall optical damping rates, respectively, the latter determining the cavity resonant linewidth ($\Delta\lambda_o = \lambda_o^2\kappa/2\pi$). The only two parameters that have been modified experimentally within this work are $P_l$ and $\lambda_l$ with a resolution below 1 μW and 1 pm, respectively. It is also worth noting the possibility of tailoring, in a much less accurate way, other parameters entering the equations, such as $\kappa_e$, $\kappa$ and the heat dissipation rate, by adjusting the fibre/sample relative alignment. In most of the situations, the long-term dynamic solution of the $\{\Delta T, N\}$ system is a stable fixed point. However, for specific combinations of $P_l$ and $\lambda_l$, the fixed point undergoes a supercritical Hopf bifurcation, transmuting into an unstable fixed point surrounded by a stable limit cycle (see Methods). The long-term solution of the $\{\Delta T, N\}$ system is then that called from now on self-pulsing (SP)[19], where the cavity resonance oscillates periodically around the laser at a frequency denoted by $\nu_{SP}$. The total time required to complete the SP limit cycle is thus $1/\nu_{SP}$, although it is not necessarily drawn at a constant pace. When the SP limit cycle is active, light within the cavity is modulated in a strongly anharmonic way, creating an optical frequency comb with multiple peaks spectrally located at integers of $\nu_{SP}$, the intensity of which decreases with frequency. An important consequence is that the intracavity radiation pressure force ($F_o$), which is proportional to $n$, is also modulated in an equivalent fashion. In a previous work[18], we demonstrated that if the $M$th harmonic of $F_o$ has a significant overlap with the linewidth of a mechanical mode with sufficient $g_{o,OM}$, the latter enters a high-amplitude, coherent regime, which fulfils the requirements to be identified as phonon lasing[4,20].

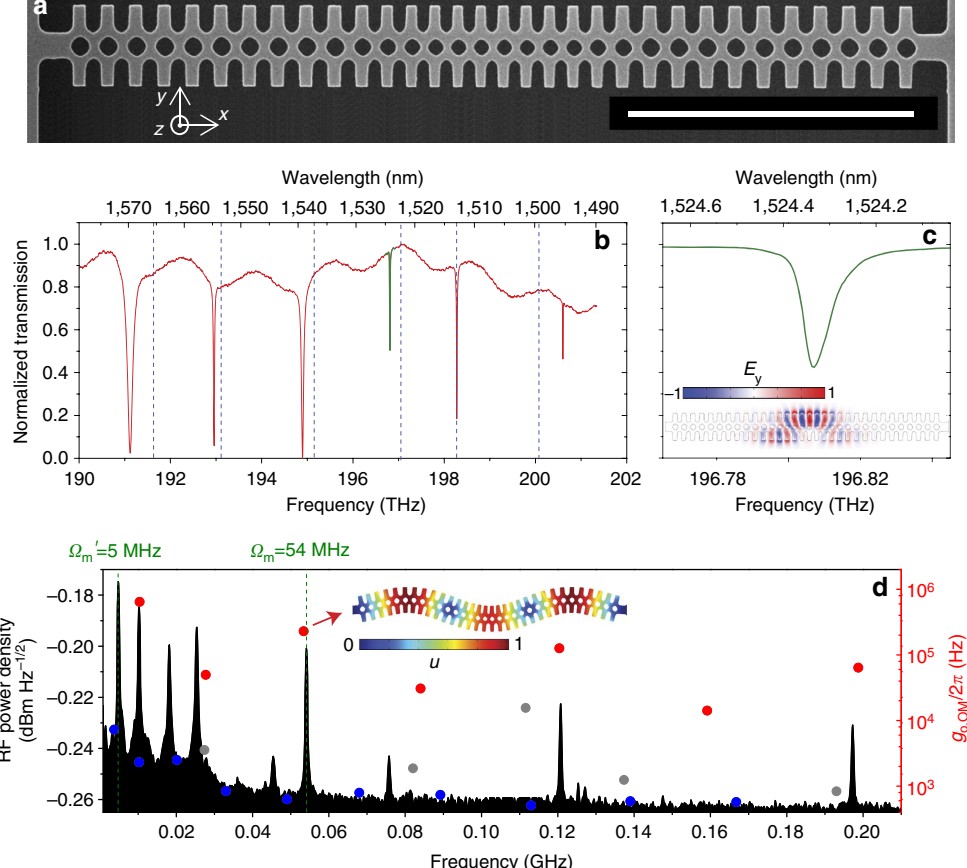

**Figure 1 | Basic properties of the OM photonic crystal.** (**a**) SEM micrograph of the fabricated structure. Scale bar, 5 µm. (**b**) Normalized transmission spectrum recorded for low laser power (<10 µW). Dashed lines indicate the spectral position of the simulated photonic modes. The specific optical resonance under study is highlighted in green. (**c**) Zoom of the optical resonance under study. Characteristic values of optical quality factor $Q_o = 2.3 \times 10^4$, extrinsic optical decay rate $\kappa_e = 1.88 \times 10^{10} \, s^{-1}$ and intrinsic optical decay rate $\kappa_i = 3.58 \times 10^{10} \, s^{-1}$ have been extracted from fitting the resonance with a Lorentzian function. The simulated $y$-component of the electric field associated to that mode is also shown (inset). (**d**) Left axis: RF spectrum obtained by exciting the cavity with the first-order optical mode (black). Right axis: Calculated values of the single-particle OM coupling rate $g_{o,OM}$ for each of the mechanical eigenmodes are reported with dots. Red and blue dots are associated to in-plane and out-of-plane flexural modes, respectively. Grey dots correspond to torsional and dilatational modes. The deformation profile ($u$) of the nanobeam associated with the mode with a mechanical eigenfrequency of $\Omega_m = 54 \, MHz$ is also shown (inset).

The transition from pure SP dynamics to the coupled SP/phonon lasing counterpart is effectively that from a two-dimensional to a four-dimensional coupled nonlinear system. Indeed, the mechanical oscillator is not just being unidirectionally driven, since the OM coupling provides an effective back coupling that forces $\nu_{SP}$ to be frequency-locked to $\nu_{SP} = \Omega_m/M$ over a wide spectral range of the laser source (see Methods). Lower $M$ values result in stronger $F_o$ and higher coupling strengths between the SP and the mechanical coherent oscillator. Once the system is found in a four-dimensional limit cycle it becomes very robust, a characteristic that can be quantified to some extent from the spectral width of the frequency-locked plateau and the low phase noise of the RF signal, which displays a main RF peak linewidth of a few kHz. These features are a direct indication of ranges of laser parameters where two stable states of the system coexist, namely the two-dimensional SP and the four-dimensional SP/phonon lasing. The existence of bistability enables the possibility of frequency jumps and hysteresis as the laser parameters are scanned. To check for these features we have studied the full RF spectra as a function of $\lambda_l$ at $P_l = 2$ mW. The experimental data were taken by sweeping $\lambda_l$ from bottom to top (Fig. 2a) and *vice versa* (Fig. 2b). As expected, $\lambda_l$-frequency jumps and $\lambda_l$-hysteretic regions appear where the system undergoes state

transitions. By singling out the position of the first RF harmonic (Fig. 2c), the $\lambda_l$-bistable regions are clearly exposed, which is also observed in the numerical simulations (see Methods).

Most of the $\lambda_l$-bistable features described above are reproduced well by numerically solving the four-dimensional coupled nonlinear system $\{\Delta T, N, u, du/dt\}$ (see Methods), where $u$ and $du/dt$ are the generalized coordinate for the displacement of the mechanical mode and its derivative, respectively. An abrupt transition between two available states occurs if the frequency difference between them ($\Delta\nu$) reaches a specific value $\Delta\nu < \Delta\nu_{up}$ ($\Delta\nu > \Delta\nu_{down}$) for an up (down) transition, where $\Delta\nu_{up} < \Delta\nu_{down}$. To unveil a hysteresis cycle, $\Delta\nu$ must fulfil the conditions for performing both an up and a down transition along the explored parameter path. In general, in our system we observe that $\Delta\nu_{up}$ and $\Delta\nu_{down}$ increase (that is, the frequency jumps are larger) with $P_l$ and for lower $M$ values, in response to an increasing of the coupling strength between the $\{\Delta T, N\}$ system and the mechanical oscillator.

Regions displaying $\lambda_l$-bistability unfold experimentally and in the numerical simulations at different spectral windows. It is worth noting that it is not possible to simultaneously operate within two of these windows because they are associated with the same photonic resonance. However, it is viable to probe photonic

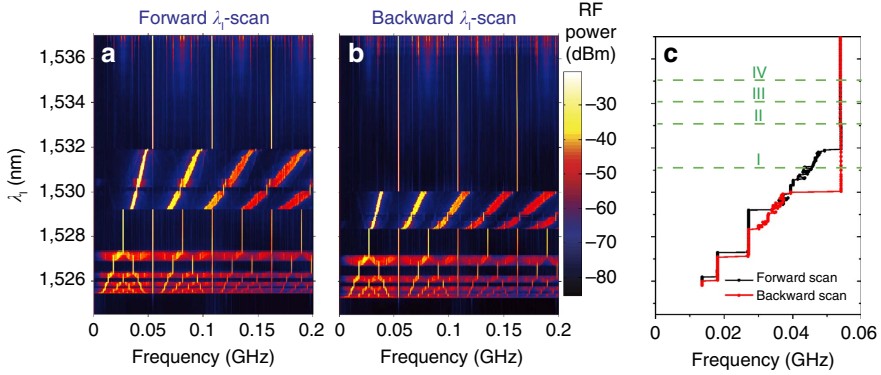

**Figure 2 | Bistability and hysteresis as a function of the laser wavelength.** (**a**,**b**) Colour contour plot of the radio-frequency spectrum as a function of the laser wavelength ($\lambda_l$) obtained at a laser power of $P_l = 2$ mW. (**a**) Corresponds to a forward $\lambda_l$-scan, while **b** corresponds to a backward $\lambda_l$-scan. (**c**) Plots the spectral position of the first radio-frequency harmonic. Horizontal dashed lines in **c** labelled with I, II, III and IV indicate the $\lambda_l$ values at which a $P_l$ scan has been performed.

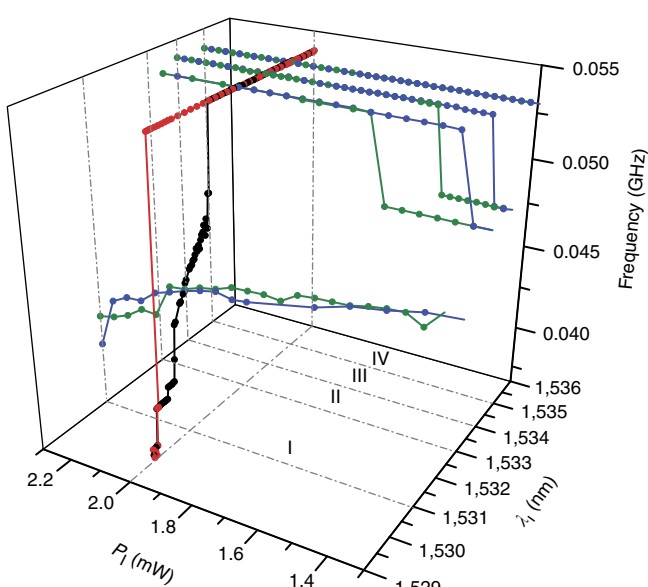

**Figure 3 | Spectral position of the first radio-frequency harmonic against the laser wavelength and power.** Black and red curves correspond to forward and backward $\lambda_l$-scans, respectively. Green and blues curves correspond to forward and backward $P_l$-scans, respectively. The dashed lines denoted by I, II, III and IV indicate four different wavelengths at which $P_l$-scans have been performed.

resonances alternative to the one being pumped, taking care not to induce further optical nonlinearities. In that case, the dynamics of the probe transmitted signals would be coherent with that of the pump, since each one just follows the common dynamics of the effective refractive indices[15].

The laser parameter space has been further explored by monitoring the system response to changes on $P_l$ for fixed values of $\lambda_l$. The analysed domain lies in the vicinity of the widest bistable region of Fig. 2c (the precise values of $\lambda_l$ under analysis are highlighted with dashed lines and called from now on I, II, III and IV, respectively). The results are reported in Fig. 3 by plotting the spectral position of the first RF harmonic peak.

Regions displaying $P_l$-bistability and $P_l$-hysteresis appear only if $\lambda_l$ is above the $\lambda_l$-frequency jump of the forward $\lambda_l$ scan. If $\lambda_l$ lies well within the $\lambda_l$-bistable region, such as in I, the system can start from either of the two stable states shown in Fig. 3 (red and black curve). However, within the studied $P_l$ range along I,

$\Delta v$ only satisfies the requirements for performing a down-transition ($\Delta v > \Delta v_{\text{down}}$), being unable to accomplish an uptransition to a $M = 1$ state. To drive the system to a $M = 1$ state, $\lambda_l$ has to be increased and, subsequently, decreased.

A $P_l$-bistable region unwraps for $\lambda_l$ values in the close vicinity of the $\lambda_l$-frequency jump, the width of which decreases with $\lambda_l$ (see green and blues curves at II and III). It fully closes at IV where the system stays robustly in the $M = 1$ plateau. Interestingly, $P_l$-bistability involves two different four-dimensional SP/phonon lasing states, namely the pervasive $\Omega_m = 54$ MHz mode with $M = 1$, and a 198 MHz mode (associated with an in-plane flexural mode having seven antinodes along the $x$-direction, see Fig. 1d and Methods) with $M = 4$, that is, the first RF peak appears at 49.5 MHz.

State switching within the $P_l$-bistable region would be enabled straightforwardly by applying power pulses/dark pulses with pulse heights dependent on the specific $\lambda_l$ in a range from a few $\mu$W to 100 s of $\mu$W. The mechanical lifetime and frequency of the modes involved would currently limit switching speeds to the megahertz range, but this could be further improved by pushing $v_{\text{SP}}$ to the gigahertz range with a smart tailoring of the heat dissipation and free-carrier recombination rates.

Regions of the laser parameters space supporting more than two stable states could also be present. Indeed, it is plausible that the $P_l$-bistable region reported in Fig. 3 could support three stable states, namely two different four-dimensional SP/phonon lasing states and two-dimensional SP. Unfortunately, we have been unable to reach all of them by just scanning $P_l$.

**Period doubling bifurcation of cycles and onset of chaos.** The dynamical solutions of the system become increasingly complex for specific ranges of $P_l$ and $\lambda_l$, giving rise to period-doubling cycles and the onset of chaos. For continuous dynamical systems the Poincaré–Bendixson theorem states that period doubling bifurcation of cycles or bifurcations towards strange attractors can only arise in three or more dimensions[1]. Therefore, this kind of bifurcations can occur, in principle, if the OM device is found in a four-dimensional SP/phonon lasing state and $P_l$ and $\lambda_l$ are varied from there. In fact, for high enough $P_l$ values, new complex dynamics appear at the sides of the plateaus. Figure 4 illustrates the most striking case, covering both sides of the $M = 1$ plateau while pumping at $P_l = 4$ mW, that is, an intracavity stored optical energy (calculated at perfect laser/cavity resonance) of 50 fJ, which is slightly lower than the energy required to disclose a chaotic regime in ref. 16 (60 fJ). In our case, the transition between the $M = 2$ and $M = 1$ plateaus does not involve pure SP

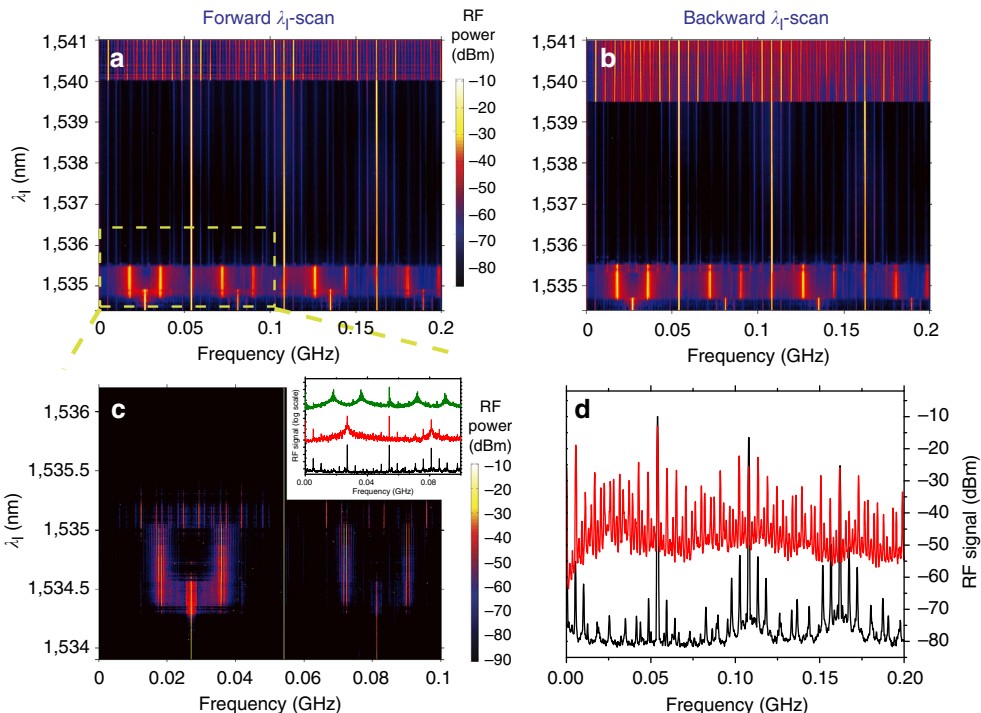

**Figure 4 | Nonlinear dynamics at high laser power.** (**a,b**) Colour contour plot of the radio-frequency spectrum as a function of the laser wavelength ($\lambda_l$) obtained at a laser power of $P_l = 4$ mW. (**a**) Correspond to a $\lambda_l$-forward scan, while panel **b** correspond to $\lambda_l$-backward scan. (**c**) Displays a zoom of the transition between $M = 2$ to $M = 1$ plateaus through subsequent period doubling bifurcations. The colour contrast has been altered with respect to **c** to highlight the period-doubling route. The inset to **c** shows radio-frequency spectra at three different laser wavelengths within the $M = 2$ to $M = 1$ transition, which are 1,534 nm (black curve), 1,534.3 nm (red curve) and 1,534.8 nm (green curve). (**d**) Shows radio-frequency spectra below (at $\lambda_l = 1,537.05$ nm, black curve) and above the bifurcation onsetting chaos ($\lambda_l = 1,540.70$ nm, red curve).

states, contrary to the observation at $P_l = 2$ mW (see Fig. 2). Along the transition, the dimension of the system is, at least, four, following a route along which the attractor undergoes subsequent period-doubling bifurcations (see lower part of Fig. 4a–c). A much broader linewidth of the $\Omega_m/2$ peak of the $M = 1$ first period doubling enables its discrimination from the $M = 2$ coherent state (red and black curves of the inset to Fig. 4c). Subsequent period doubling bifurcations appear before entering into the $M = 1$ coherent regime, with its peak intensity decreasing as the period doubling level increases. On the other side of the $M = 1$ plateau the system suffers another bifurcation in which it abruptly enters into a chaotic regime within a strange attractor volume in the phase space. The RF spectra below and above that bifurcation are shown in Fig. 4d (black and red curves, respectively). A chaotic behaviour is also foreseen by our numerical simulations for high enough $\lambda_l$ values at high $P_l$ (see Methods and Supplementary Figs 9–11). The simplest version of our model, which considers a single mechanical mode at $\Omega_m$, predicts: the onset of chaos in the $\Delta T$ and $N$ space by a smooth period-doubling route and a coherent high-amplitude oscillation of the mechanical mode. This would lead to a flat broadband RF spectrum with sharp peaks only at the frequencies of the $\Omega_m$ harmonics. In the experiment, the onset of chaos is abrupt and leads to a broadband RF spectrum as well, but with a much richer structure of sharp peaks (red curve of Fig. 4d). The most intense ones are associated to the $\Omega_m$ mode (main peak plus harmonics) lasing in the plateau, similarly to the numerical simulation (see Methods and Supplementary Fig. 10). The remaining peaks are ascribed to the activation of the fundamental out-of-plane flexural mode described before ($\Omega'_m = 5$ MHz), providing peaks at $\Omega'_m$, at its harmonics and at sidebands of the $\Omega_m$-associated peaks. On the basis of these experimental results, and as predicted,

we conclude that a chaotic regime is established by the SP/single mechanical mode lasing state, thus creating a broad band $F_o$. However, this is not the stable solution of the system since, in response to $F_o$, other mechanical modes displaying large enough $g_{o,OM}$ can be activated. The nonlinear system thus increases its dimension by a number equal to twice the number of mechanical modes involved. We have confirmed this hypothesis by including a second harmonic oscillator at $\Omega'_m$ in our model (see Methods and Supplementary Fig. 11) driven by $F_o$, predicting that the chaotic dynamics are only present in $\Delta T$ and $N$, while the two mechanical modes oscillate coherently at incommensurate frequencies. It is worth noting that the only additional mode to be amplified is an out-of-plane flexural one having a large $g_{o,OM}$ value, while the other in-plane ones, although having large $g_{o,OM}$ values as well, are probably damped because of mechanical mode competition[21].

There are other two remarkable features reported in Fig. 4a,c: the absence of hysteresis in the transition between $M = 2$ and $M = 1$ and the hysteresis cycle involving chaos and the SP/single mode lasing limit cycle. The former is in contrast with what is reported in Fig. 2, leading us to conclude that the hysteresis cycles reported above only appear when the transition is through two-dimensional SP, that is, at low $P_l$. The second feature is an indication of two types of coexisting attractors: fixed points and strange attractors. It also means that a large enough instantaneous perturbation can knock the coherent oscillation into chaos or *vice versa*. The complete insight of the latter bistability exceeds the scope of this work, but we suggest it is a direct consequence of the coupling of the two mechanical modes through $n$.

The time evolution of the transmitted signal below and above the abrupt bifurcation delimiting the onset of chaos is reported in Fig. 5. As expected, above the bifurcation, the signal is aperiodic

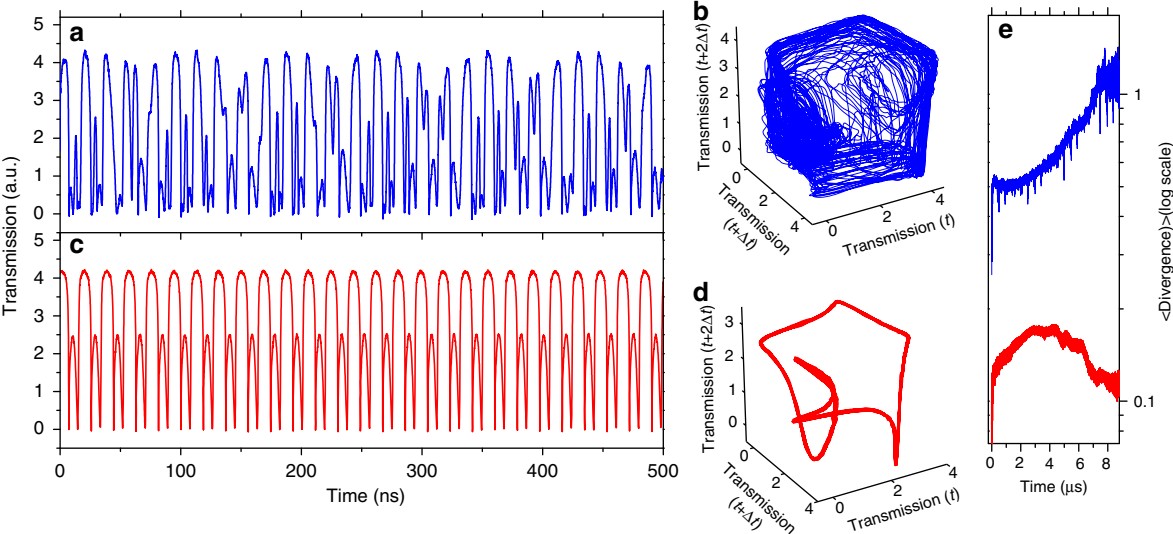

**Figure 5 | Time evolution above and below the bifurcation onsetting chaos.** Time series of the transmitted signal above and below the bifurcation onsetting chaos (**a,c**, respectively). Reconstruction of the embedding state in a three-dimensional projection as extracted from the temporal signals (**b,d**). (**e**) Time evolution of the divergence (in log scale) extracted from the experimental signals by applying the Rosenstein algorithm. The input parameters are: $m = 8$; delay = 10 ns; mean period = 20 ns.

while below the bifurcation the signal is coherent. Aiming to reconstruct the embedding state in a projection on a three-dimensional space, in Fig. 5b,d we plot the transmitted signal against itself delayed 10 ns and itself delayed 20 ns. A limit cycle is reconstructed for the coherent case, while the described trajectory in the chaotic case tends to fill a restricted volume. It is worth noting that, by weakly probing another photonic resonance in addition to the one being pumped, the chaotic dynamics of the pump signal will be transferred to the probe, since each one just follows the common dynamics of the effective refractive indices[15].

Finally, we applied the Rosenstein algorithm[22] to 10-µs-long transmission registers taken with a temporal resolution of 10 ps (see Methods and Supplementary Fig. 12). The output is the temporal evolution of the average divergence of initially close state-space trajectories (Fig. 5e). The applied algorithm starts converging for an embedding dimension $m \geq 6$. The latter must be at least equal to the dimension of the system[22], which agrees with the dimension of our model considering two mechanical modes. Largest Lyapunov exponents (LLE) quantify the exponential behaviour of the divergence and estimate the amount of chaos in the system. The sensitive dependence on initial conditions, quantified by LLE, preclude long-term predictions, but it promises improved short-term predictions[23]. From the exponential fit of the blue curve of Fig. 5e we extract an LLE value of $1.3 \times 10^5 \, \mathrm{s}^{-1}$. This value is about one order of magnitude lower than $\Omega'_m$ and two orders of magnitude lower than $\Omega_m$, which are the typical frequencies of the system. Obviously, the divergence growing trend saturates at some point, since trajectories cannot go any further than the scale of the attractor volume, which is of the order of the amplitude of the transmission oscillations. To check for the validity of the used parameters, we applied the same algorithm to the coherent signal (red curve in Fig. 5e), the result being that the trajectories stay parallel at a divergence value that is almost an order of magnitude smaller than in the chaotic case.

An estimation of the LLE in OM systems has been realized in refs 15,16. In the former reference, its value is provided without units, preventing any comparison with our work. In the case of ref. 16, the LLE exceeds by three orders of magnitude the one reported here and is much larger than the typical frequencies of the system. In that work, the LLE is calculated using the Grassberger–Procaccia method proposed in ref. 24, which, as stated in a later ref. 22, is sensitive to variations of its input parameters (number of data points, embedding dimension, reconstruction delay) and it is usually unreliable except for long, noise-free time series. In our case, we apply the Rosenstein algorithm, which is robust where Grassberger–Procaccia is not and is widely acknowledge to be accurate because it takes advantage of all the available data and works well with small data sets.

## Discussion

We have reported a rich set of complex dynamical solutions, including chaos, experimentally observed in a single and compact Si-based OM nanobeam, optically pumped with a continuous-wave, low-power laser source. The reported features are very robust and stable for hours, in spite of being achieved at atmospheric conditions of temperature and pressure.

We have provided a glimpse of the dynamical map of the system by exploring the laser parameters (power and wavelength) in different senses. In particular, at low laser powers, we have demonstrated hysteresis and bistability of two kinds involving: a two-dimensional SP and a four-dimensional SP/phonon lasing state and two four-dimensional SP/phonon lasing states involving two mechanical modes. The potential application, for instance for OM memories, of the reported $P_l$-bistability is probably higher than the $\lambda_l$-bistability, since state switching would be achieved by applying power pulses.

At high laser powers, we have observed that the transition between two consecutive four-dimensional SP/phonon lasing states is through a period-doubling route. In addition, we have demonstrated the onset of a six-dimensional chaos displaying bistability and hysteresis with a four-dimensional SP/phonon lasing state. We envision the use of many of those chaotic optical sources on an SOI chip in a multichannel approach. Each device will eventually show chaotic dynamics that are dependent on the specific geometry of the OM crystal and of the parameters of the laser used to pump the optical resonance, effectively behaving as a multichannel chaotic source that could be multiplexed into a single optical fibre.

Besides the specific applications that some of the previous features may find, namely for memories, switches, sensing or secure data transmissions, the straightforward access to a large part of the parameter space makes natural the exploitation of this device as a test bed for investigating uncommon complex nonlinear dynamics such as static and dynamic bifurcations, different routes to chaos and so on. The nonlinear system can easily increase its complexity, for instance, by pumping several optical modes simultaneously or by weakly coupling several self-sustained OM devices, from pairs to networks. Thus, we propose our device as a versatile building block to develop mechanical neurocomputational architectures[25].

It is worth noting that a recent ref. 16 reports on a two-dimensional integrated geometry showing similarities to our system in terms of some of the observed dynamical solutions. In addition to the claim of chaotic dynamics, which has been discussed in comparison with our findings in the previous section, ref. 16 reports pure SP and several phonon lasing regimes, each one activated using different harmonics of the optical force. Those latter features are in agreement with what was reported by us previously in ref. 18. A comprehensive comparison of the works of both groups, including an analysis of the input parameters at which those regimes are triggered, is included in the Supplementary Discussion.

## Methods

**OM crystal geometry.** The investigated device is an OM photonic crystal, the unit cell of which contains a hole in the middle and two symmetric stubs on the sides (see Supplementary Fig. 1). The peculiarity of this geometry is a full phononic bandgap $\sim 4\,$GHz (ref. 26). The investigation of high-frequency mechanical modes is reported elsewhere[27]. We describe here the geometrical parameters of our device. In a defect region consisting of 12 central cells the pitch ($a$), the radius of the hole ($r$) and the stubs length ($d$) are decreased in a quadratic way towards the centre. The maximum reduction of the parameters is denoted by $\Gamma$. A 10 period mirror is included at both sides of the defect region. The nominal geometrical values of the cells of the mirror are $a = 500\,$nm, $r = 150\,$nm and $d = 250\,$nm. The total number of cells is 32 and the whole device length is $\sim 15\,\mu$m. We have fabricated a set of devices in which $\Gamma$ has been varied from $\Gamma = 64\%$ to $\Gamma = 83\%$ of the original values. All the results presented in this work correspond to the structure with $\Gamma = 83\%$, but the same effects have been observed for other values of $\Gamma$.

**OM crystal fabrication.** The devices were fabricated in standard SOI Soitec wafers with silicon layer thickness of 220 nm (resistivity $\rho \sim 1$–$10\,\Omega\,$cm$^{-1}$, $p$-doping of $\sim 10^{15}\,$cm$^{-3}$). The pattern was written by electron beam lithography in a 100-nm-thick poly-methyl-methacrylate resist film and transferred into silicon by reactive ion etching. Application of buffered hydrofluoric acid removed the buried oxide layer and released the beam structures.

**Measurements.** The experiments are performed in a standard set-up for characterizing optical and mechanical properties of OM devices. A tunable infrared laser covering the spectral range between 1,460–1,580 nm is connected to a tapered fibre. The polarization state of the light entering the tapered region is set with a polarization controller. The thinnest part of the tapered fibre is placed parallel to the OM photonic crystal, in contact with an edge of the etched frame (top right photo of Supplementary Fig. 2a,b). The gap between the fibre and the structure is about 200 nm. A polarization analyser is placed after the tapered fibre region. The long tail of the evanescent field and the relatively high spatial resolution ($\sim 5\,\mu$m$^2$) of the tapered fibre locally excited the resonant optical modes of the OM photonic crystal. Once on resonance, the mechanical motion activated by the thermal Langevin force causes the transmitted intensity to be modulated around the static value (Supplementary Fig. 2c). To check for the presence of a radio-frequency modulation of the transmitted signal an InGaAs fast photoreceiver with a bandwidth of 12 GHz was used. The radio-frequency voltage is connected to the 50 $\Omega$ input impedance of a signal analyser with a bandwidth of 13.5 GHz. All the measurements were performed in an antivibration cage at atmospheric conditions of air pressure and temperature.

**Tapered fibre characteristics and fabrication.** The experiments are carried out with tapered optical fibres having diameters in the smallest section of $\sim 1.8\,\mu$m (Supplementary Fig. 3a), which is commensurate with the wavelength of interest ($\sim 1.5\,\mu$m) and ensures an evanescent field tail of several hundreds of nanometres. For the fibre fabrication, we used a home-made setup in which an SMF-28 optical fibre is stretched in a controlled way using two motorized stages. The central part

of the fibre is placed in a microheater where the temperature is about 1,180 °C (ref. 27). The fibre transmission at a wavelength of 1.5 μm is monitored during the pulling procedure (Supplementary Fig. 3b). The signal is subjected to a short time fast Fourier transform algorithm, so that the frequency components associated with inference between different supported modes are measured (Supplementary Fig. 3c). The single-mode configuration is achieved when all those frequency components disappear (Supplementary Fig. 3d). Using two rotating fibre clamps, the tapered fibre is twisted twice around itself. The two fibre ends are gently brought closer over several hundreds of micrometres so that a looped structure forms in the tapered region. The two fibre ends are afterwards pulled apart to reduce the loop size down to a few tens of microns. In that process, the two parts of the fibre at the loop closing point (upper part of the loop on the top left photo Supplementary Fig. 3) slide smoothly in opposite senses. The micro-looped shape provides functionalities similar to those of dimpled fibres. For the fibre loop photonic structure, the dispersion relation is linear and the group refractive index ($n_{\mathrm{g}}$) is equivalent to the effective refractive index ($n_{\mathrm{eff}}$). We have calculated $n_{\mathrm{eff}}$ using the beam propagation method taking the material refractive index of the cladding of the initial fibre to be $n = 1.468$ at $\lambda_{\mathrm{l}} = 1{,}515\,$nm, resulting in a value of $n_{\mathrm{eff}} = 1.373$.

**FEM simulations.** COMSOL FEM simulations of a complete structure are used to determine the fundamental cavity mode frequencies, effective masses of the mechanical eigenmodes and single-particle OM coupling rates ($g_{\mathrm{o,OM}}$). To model the OM crystal system, the contour of the of the as-fabricated structure SEM image was extracted graphically and then imported in the FEM solver (Supplementary Fig. 4). Among the many optical and mechanical modes supported by the OM crystal, there are specifically one optical and three mechanical that are discussed in the main text. Those are the third optical mode appearing at 197 THz (Supplementary Fig. 5a) and the mechanical modes appearing at $\Omega'_m = 5\,$MHz, $\Omega_m = 54$ and 198 MHz (Supplementary Fig. 5b–d, respectively), whose effective masses are $12 \times 10^{-12}$, $13 \times 10^{-12}$ and $11 \times 10^{-12}\,$g, respectively. Single-particle OM coupling rates between optical and mechanical modes are estimated by taking into account both the photoelastic (PE) and the moving interface (MI) effects[28–30]. The PE effect is a result of the acoustic strain within bulk silicon, while the MI mechanism comes from the dielectric permittivity variation at the boundaries associated with the deformation. The calculation of the MI coupling coefficient $g_{\mathrm{MI}}$ is performed using the integral given by Johnson et al.[28]:

$$g_{\mathrm{MI}} = -\frac{\pi \lambda_{\mathrm{r}}}{c} \frac{\oint (\mathbf{Q} \cdot \mathbf{n})\left(\Delta\varepsilon \mathbf{E}_{\parallel}^2 - \Delta\varepsilon^{-1}\mathbf{D}_{\perp}^2\right)}{\int \mathbf{E} \cdot \mathbf{D}\,\mathrm{d}V} \sqrt{\frac{\hbar}{2m_{\mathrm{eff}}\Omega_m}}, \qquad (2)$$

where $\mathbf{Q}$ is the normalized displacement (max$|\mathbf{Q}| = 1$), $\mathbf{n}$ is the normal at the boundary (pointing outward), $\mathbf{E}$ is the electric field and $\mathbf{D}$ the electric displacement field. The dielectric permittivity is denoted by $\varepsilon$, $\Delta\varepsilon = \varepsilon_{\mathrm{silicon}} - \varepsilon_{\mathrm{air}}$, $\Delta\varepsilon^{-1} = \varepsilon_{\mathrm{silicon}}^{-1} - \varepsilon_{\mathrm{air}}^{-1}$. The speed of light in vacuum is denoted by $c$, is the reduced Planck constant, $m_{\mathrm{eff}}$ is the effective mass of the mechanical mode and $\Omega_m$ is the mechanical mode eigenfrequency, so that is the zero-point motion of the resonator. A similar result can be derived for the PE contribution[29,30]:

$$g_{\mathrm{PE}} = -\frac{\pi \lambda_{\mathrm{r}}}{c} \frac{\langle \mathbf{E}|\delta\varepsilon|\mathbf{E}\rangle}{\int \mathbf{E} \cdot \mathbf{D}\,\mathrm{d}V} \sqrt{\frac{\hbar}{2m_{\mathrm{eff}}\Omega_m}}, \qquad (3)$$

where $\delta_{ij} = \varepsilon_{\mathrm{air}} n^4 p_{ijkl} S_{kl}$, being $p_{ijkl}$ the PE tensor components, $n$ the refractive index of silicon and $S_{kl}$ the strain tensor components. The addition of both contributions results in the overall single-particle OM coupling rate:

$$g_{\mathrm{o,OM}} = g_{\mathrm{MI}} + g_{\mathrm{PE}}, \qquad (4)$$

It is worth noting that the string-like modes studied in this work have very low PE contribution in comparison with the MI counterpart. From the analysis of the MI surface density (the integrand of equation (2)) of the $\Omega'_m = 5\,$MHz mode (Supplementary Fig. 6) it is possible to conclude that the contribution provided by the top and bottom Si–air interfaces are both quite large, very similar in absolute value (about $g_{\mathrm{MI}}/2\pi = 1\,$MHz) but with opposite signs. As a consequence of the latter, a rather low value of $g_{\mathrm{MI}}$, namely $g_{\mathrm{MI}}/2\pi = 7\,$kHz, is calculated. However, geometric inhomogeneities in the real structure along the $z$ axis may break that axis, unbalancing the contributions from the top and bottom Si–air interfaces. On the contrary, the MI surface density to the $\Omega_m = 54\,$MHz mode (see Supplementary Fig. 7) allows concluding that the main contribution comes from the lateral Si–air interface. The outer stub surface and the inner hole surface contribute with different signs, but the overall value is rather large, namely $g_{\mathrm{MI}}/2\pi = 650\,$kHz.

**Self-pulsing and phonon lasing.** This section addresses the general mechanism of SP limit cycles in optical resonators and the specific case of phonon lasing driven by SP occurring in OM cavities. An extended discussion of these phenomena can be found in ref. 18.

We consider the two main sources of optical nonlinearities in silicon-based cavities, which are the thermo-optic effect and free-carrier dispersion[19]. In the former effect, there is an associated redshift of the optical resonance as a function of the effective temperature of the cavity ($\Delta T$), while in the latter effect the spectral

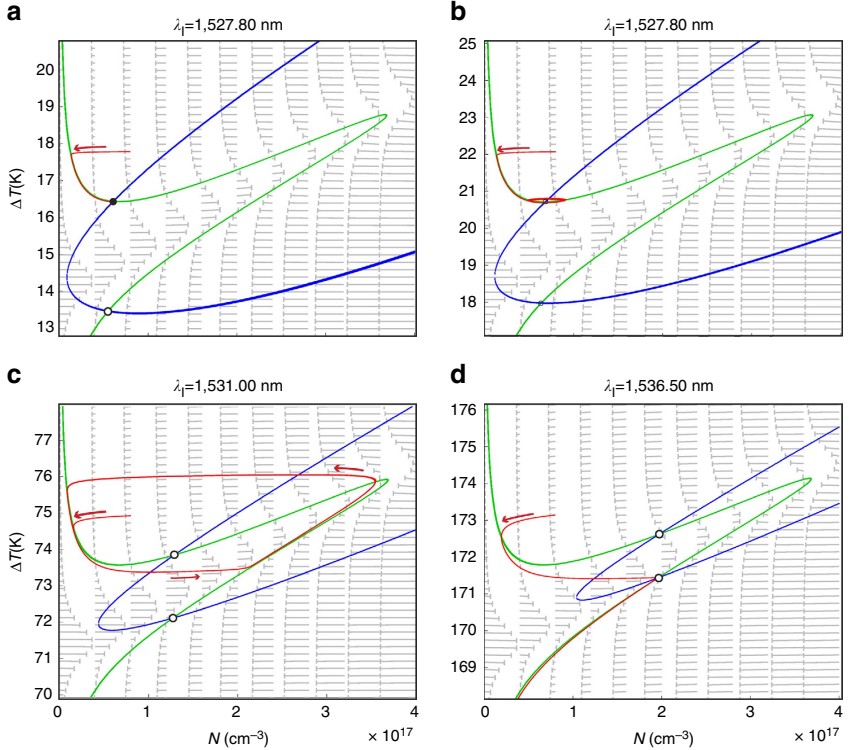

**Figure 6 | Phase portraits for different laser wavelengths.** Phase portraits as a function of the free-carrier density ($N$) and the average temperature increase ($\Delta T$) for $\lambda_I$ below the supercritical Hopf bifurcation (**a**), at the bifurcation (**b**). Above the bifurcation we show the SP limit-cycle regime (**c**) and the situation in which the limit cycle is wiped out due to the interception with a second unstable fixed point (**d**). Solid (open) circles indicate stable (unstable) fixed points. The nullclines of equations (5) and (6) are plotted in green ($\dot{N} = 0$) and blue ($\dot{\Delta T} = 0$), respectively, while the simulated trajectories are in red. Grey arrows depict the temporal derivatives of $\Delta T$ and $N$ in each point of the phase space, indicating the direction towards which the system would evolve.

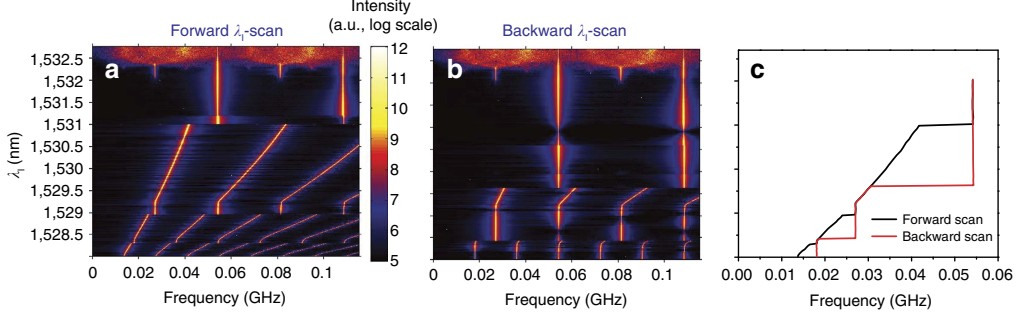

**Figure 7 | Simulated bistability and hysteresis as a function of the laser wavelength.** (**a,b**) Colour contour plot of the simulated radio-frequency spectrum of the transmitted signal as a function of the laser wavelength ($\lambda_I$) obtained for a maximum intracavity photon number of $n_o = 10^5$. (**a**) Corresponds to a forward $\lambda_I$-scan, while **b** corresponds to a backward $\lambda_I$-scan. (**c**) Plots the spectral position of the first harmonic peak.

shift is towards the blue as a function of the density of free carriers ($N$). Since $\Delta T$ is directly related with nonradiative recombinations of free carriers, the differential equations governing both optical nonlinear effects are linked, and can be expressed in the following terms:

$$\dot{N} = -\frac{1}{\tau_{FC}} N + \beta \left( \frac{hc^3}{n_r^2 \lambda_o V_o^2} n^2 \right), \qquad (5)$$

$$\dot{\Delta T} = -\frac{1}{\tau_T} \Delta T + \alpha_{FC} N n, \qquad (6)$$

where $V_o$ is the optical mode volume and $n_r$ is the refractive index. Two-photon absorption is considered as the main free-carrier generation mechanism in the free-carrier density differential equation (equation (5)), where $\beta$ is the tabulated two-photon absorption coefficient. Equation (5) also includes a surface recombination term governed by a characteristic lifetime $\tau_{FC}$. We consider free-carrier absorption as the main heating mechanism in equation (6), where $\alpha_{FC}$ is defined as the rate of temperature increase per photon and unit free-carrier density. The heat dissipated

to the surroundings of the cavity volume is quantified by the characteristic lifetime $\tau_T$. Importantly, $\dot{\lambda}_I \approx \lambda_o - \frac{\partial \lambda_r}{\partial N} N + \frac{\partial \lambda_r}{\partial \Delta T} \Delta T$ is the cavity resonant wavelength including first-order nonlinear effects. Since $\kappa^{-1}$ is much smaller than $\tau_T$ and $\tau_{FC}$, it is possible to consider that the response of the optical cavity to free-carrier dispersion and thermo-optic-induced refractive index changes is adiabatic. Experimentally we have access to both laser parameters ($P_I$ and $\lambda_I$), which are included in the dynamic equations implicitly in $n$, thus impacting the generation rates. Figure 6 illustrates the four different solutions that equations (5) and (6) may show depending on $\lambda_I$ (please note that we have kept $P_I$ fixed at 2 mW). For $\lambda_I$ values close to $\lambda_o$ (Fig. 6a), the system displays two fixed points, one stable (solid circle) that attracts neighbouring trajectories and one unstable (empty circle) that repels trajectories within its vicinity. In the experiment, for $\lambda_I$ values close to $\lambda_o$, the system is found at the stable fixed point. Thermal or mechanical disturbances are confined in the attractor phase surface associated with the fixed point so that $\{N, \Delta T\}$ is found in its close vicinity. If driven away from that point, the system settles down to equilibrium through exponentially damped oscillations. By gradually increasing $\lambda_I$ the system eventually experiences a supercritical Hopf bifurcation (Fig. 6b). Just at the bifurcation the fixed point is still stable, although it

is a weak one, and the system decays to it algebraically if found in its vicinity. Slightly above the bifurcation, the stable fixed point transforms into an unstable fixed point surrounded by a stable limit cycle (Fig. 6c). If the system is driven away from the limit cycle, it will rapidly converge to it again. This limit cycle is defined as the SP limit cycle and is the stable dynamical solution of equations (5) and (6) for a wide range of laser parameters. It was first demonstrated experimentally by Johnson et al.[19] in a silicon microdisk. In the SP limit-cycle regime, the pair of values {$N, \Delta T$} oscillate periodically with a frequency denoted by $\upsilon_{SP}$, although the closed trajectory is not described at a constant pace. In the experiment, the SP limit cycle induces an anharmonic periodic modulation of the transmitted optical signal and, hence, of $n$. By further increasing $\lambda_l$ the SP increases $\upsilon_{SP}$, which is a consequence of the higher average temperature of the system. Another important effect of increasing $\lambda_l$ is that the second unstable fixed point gets closer to the limit-cycle trajectory. The SP limit cycle is wiped out when that unstable fixed point intercepts the limit-cycle trajectory (Fig. 6d), deflecting out its path and driving the system down to a situation in which $\Delta T = 0$, where the cavity resonance is far away from $\lambda_l$.

Hereafter, we consider the case of an OM cavity with mechanical degrees of freedom that can be driven by optical forces. When the SP limit cycle solution is active the radiation pressure force ($F_o$) is modulated in the same way as $n$, since they are related in a linear way, that is, $F_o = \hbar g_{o,OM} n$. The differential equation governing the generalized coordinate for the displacement of a mechanical mode ($u$) is that of a damped linear harmonic oscillator driven by $F_o$:

$$m_{eff}\ddot{u} + m_{eff}\frac{\Omega_m}{Q_{m,i}}\dot{u} + k_{eff}u = F_o, \qquad (7)$$

where $m_{eff}$, $k_{eff}$ and $\Omega_m$ are its effective mass, spring constant and eigenfrequency, respectively. The nonlinear resonance position includes now the effect of the mechanical motion when evaluating $n$, that is, $\lambda_l \approx \lambda_o - \frac{\partial \lambda_r}{\partial N}N + \frac{\partial \lambda_r}{\partial \Delta T}\Delta T + \frac{\lambda_o^2 g_{o,OM}}{2\pi c}u$. Importantly, the response of $n$ to $u$ is also adiabatic since $\kappa$ is a few orders of magnitude smaller than $\Omega_m$. Equations (5)–(7) describe the dynamics of a four-dimensional {$N, \Delta T, u, u'$} nonlinear system that is coupled through $n$. One of the consequences of the coupling of the two systems is that self-sustained mechanical motion is achieved if one of the low harmonics of the SP main peak at $\upsilon_{SP}$ is resonant with the mechanical oscillations ($M\upsilon_{SP} = \Omega_m$, where $M \in \mathbb{Z}$)(Supplementary Fig. 8). The experimental features of this coupled system can be well reproduced with the couple set of equations (5)–(7), the only free parameters being $\tau_{FC}$ and $\tau_T$ (Supplementary Fig. 8b). Those are extracted by fitting the temporal trace of the transmitted signal in an unlocked configuration using equations (5) and 6)[18].

It is worth noting that the phonon laser limit cycle is finally wiped out in a similar fashion as in the pure SP case, that is, the second unstable fixed point intercepts the limit-cycle trajectory and drives the cavity out of resonance.

### Modelling of hysteresis and chaos with one mechanical mode.
We have numerically solved the system of four nonlinear equations composed by equations (5)–(7). The simulations are done sequentially for increasing (decreasing) values of $\lambda_l$ using initial conditions provided by one point {$N, \Delta T, u, u'$} of the stable state calculated for the preceding $\lambda_l$ value. The obtained results are summarized in Fig. 7, where we plot the fast Fourier transform of the transmitted signal. The model predicts, on the one hand, bistability and hysteresis between two-dimensional SP and four-dimensional SP/phonon lasing, thus displaying some of the features reported in the main text. The model also accounts for the onsetting of chaos on ($N, \Delta T$), while the mechanical mode oscillates coherently (sharp peaks appear at $\Omega_m = 54$ MHz and its harmonics). However, the predicted route is much smoother than the experimental and through subsequent period doubling bifurcations instead of through an abrupt transition. It is worth noting that, again here, the period doubling dynamics is only present in ($N, \Delta T$), while the mechanical mode oscillates in a coherent fashion. Simulated trajectories associated to a four-dimensional SP/phonon lasing and to chaos are shown in Supplementary Fig. 9. The chaotic dynamics are robust along a wide spectral range, and are only wiped out when the second unstable fixed point shown in Fig. 6 gets close enough to the strange attractor volume, effectively deflecting down the trajectory of the system (like in Fig. 6d). The cavity then relaxes down to the initial cold situation at $\Delta T = 0$, where the cavity resonance is far away from the laser line. In the measurements shown in Fig. 4 we have not reached that condition, since by doing so we would have lost the possibility of performing the back-scan measurement of Fig. 4b. The model described by equations (5)–(7) captures most of the features described along the main text. However, it does not incorporate three main experimental observations, namely: the period doubling bifurcation route between $M = 2$ and $M = 1$, the presence of bistability and hysteresis between four-dimensional SP/phonon lasing and chaos and the specific peak structure of the chaotic RF signal. The first observation remains a subject for further studies, since we predict much smaller high-order plateaus that what measured experimentally, which eventually prevents reaching a similar situation to that reported in Fig. 4c. In the following section we address the former observations.

### Modelling of chaos with two mechanical modes.
As explained briefly in the main text, most of the sharp peaks that are present in the chaotic RF spectrum

reported in Fig. 4 are associated to the $\Omega_m$ mode (main peak + harmonics) that was lasing in the plateau, similarly to what is reported in the simulation of the previous section. The remaining peaks are associated with the activation of the fundamental out-of-plane flexural mode described before ($\Omega'_m = 5$ MHz), providing peaks at $\Omega'_m$, at its harmonics and at sidebands of the $\Omega_m$-associated peaks. To account for that mode in the model, we have included a second harmonic oscillator in addition to equations (5)–(7). The force driving the second oscillator is very likely radiation pressure ($F'_o$), because that particular mode displays the stronger transduced signal below the threshold for SP, where thermal Langevin forces are dominant. We rule out pure mechanical nonlinear interactions because, if they were to be accounted for, other mechanical modes should also be active when the system is in a plateau, which is not the case. The additional equation thus reads:

$$m_{eff,2}\ddot{u}_2 + m_{eff,2}\frac{\Omega'_m}{Q_{m,i,2}}\dot{u}_2 + k_{eff,2}u_2 = F'_o. \qquad (8)$$

The $g_{o,OM}$ value for this specific mode has been taken by scaling the $g_{o,OM}$ value corresponding to the $\Omega_m = 54$ MHz mode by the RF signal ratio between the two modes. The obtained results are very similar to those of the previous section. Within the $M = 1$ plateau, the second harmonic oscillator is off (see Supplementary Fig. 10b). However, when the system enters into the chaotic regime, it gets activated as a consequence of the chaotic dynamics of $F'_o$. Through OM coupling its oscillations modulate $\lambda_r$. As a result of the latter, the fast Fourier transform of the transmitted signal displays sharp peaks at $\Omega_m$ and $\Omega'_m$ (see Supplementary Fig. 10a). It is also possible to distinguish sidebands at $\Omega_m \pm \Omega'_m$. The first harmonic oscillator oscillates coherently also in the chaotic regime, which again is only present in ($N, \Delta T$). In Supplementary Fig. 11, we report temporal series spanning 1 μs of the different magnitudes described by the nonlinear system of equations (5)–(7) and the resulting transmitted signal.

### Application of the Rosenstein algorithm.
We have used the Rosenstein method for calculating the LLE from our experimental series[22]. The output is the temporal evolution of the average divergence of initially close state-space trajectories. The method follows directly from the definition of LLE and is accurate because it uses the whole available set of data. Although it is suitable for relatively small data sets, the temporal series has to be well resolved and span over a temporal range that should be several times greater than the predictability horizon of the system. Otherwise, the results will not be trustworthy and the extracted LLE values will be orders of magnitude greater than the real ones, that is, the trajectories coming from as-close-as-possible initial states will almost immediately diverge up to the scale of the attractor volume. Indeed, this was our case when applying the Rosenstein algorithm to short chaotic time series (few hundreds of nanoseconds), since, given a reference state on the time series, it was not possible to find a good as-close-as-possible state in the remaining set of data. Another critical input value is the embedding dimension of the system ($m$). It is imperative to evaluate the algorithm using different values for $m$. In Supplementary Fig. 12 it is apparent that satisfactory results are obtained only when $m$ is, at least, six, which is indeed the dimension of the system. This is due to the fact that chaotic systems are effectively stochastic when embedded in a phase space that is too small to accommodate the true dynamics. Finally, it is worth noting that the superposition of two sinusoids (for instance, two harmonic oscillators) with incommensurate frequencies is a non-chaotic system, but is quasiperiodic and deterministic. Although the time series would look chaotic, applying the Rosenstein algorithm to such a signal would lead to a flat divergence curve[22], leading to a null LLE. In our case the divergence is clearly growing with time, which is a clear signature of chaos.

### Data availability.
All data used in this study are available from the corresponding author on request.

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

## Acknowledgements

This work was supported by the European Comission project PHENOMEN (H2020-EU-713450), the Spanish Severo Ochoa Excellence program and the MINECO project PHENTOM (FIS2015-70862-P). DNU, PDG and MFC gratefully acknowledge the support of a Ramón y Cajal postdoctoral fellowship (RYC-2014-15392), a Beatriu de Pinos postdoctoral fellowship (BP-DGR 2015 (B) and a Severo Ochoa studentship, respectively. We would like to acknowledge Jose C. Sabina de Lis, J.M. Plata Suárez, A. Trifonova and C. Masoller for fruitful discussions.

## Author contributions

D.N.-U. performed the experiments. D.N.-U. and N.E.C. developed the model and analysed the data. M.F.C., P.D.G., M. S., F.A. and C.M.S.-T. participated in the experiments at different stages. A.G. and A.M. fabricated the samples. All authors contributed to the interpretation of the results and writing of the manuscript.

## Additional information

**Competing interests:** The authors declare no competing financial interests.

**Publisher's note**: 

