## [Peer Review File · Nature Communications]

Reviewers' comments:

Reviewer #1 (Remarks to the Author):

The paper by Navarro-Urrios et al. reports measurements on an integrated one-dimensional optomechanical (beam) cavity composed of a cleverly designed Si-based photonic nanostructure with a lattice constant reduced towards the center of the beam. The authors analyze the dynamics of the system being mainly determined by optical-linearities due to thermo-optic, free-carrier dispersion and opto-mechanical coupling effects. Modifying exclusively the laser's power and wavelength they look for bi- and higher-dimensional limit cycles and, most interestingly, for a period doubling bifurcation route to chaos in optomechanical systems. Those states are precisely identified and characterized by RF spectra, see the wave-length forward and backward scans in Figs. 2 and 4 for moderate and high laser power, respectively. The interpretation of the results (and especially the onset of chaos) is flanked and supported by numerical simulations.

The time evolution of the transmitted signal below and above the bifurcation onset of chaos (Fig. 5) is also interesting-looking: the periodic respectively aperiodic behavior is clearly demonstrated and the largest Lyapunov Exponent is extracted (in order to estimate for the amount of chaos).

In my opinion the study presented is a nice piece of experimental work on a really contemporary issue with potential applications in the rapidly developing field of photonics/phononics. The only point of criticism at this manuscript is that the authors have widely ignored previous theoretical work on non-linear dynamics and chaos in optomechanical systems. For example, the route to chaos in optomechanics considered in this paper has been worked out for a paradigmatic optomechanical model system by Bakemeier et al in a recent PRL (114, 013601 (2015)) as well as optomechanical multistability and the existence of self-sustained oscillations (EPL 113, 64002 (2016)). The authors

should definitely give credit to these papers and related work in the introduction and might relate their findings to these predictions (at least qualitatively) when discussing their data, before I can approve this paper for publication in Nature Communications.

Reviewer #2 (Remarks to the Author):

The manuscript by Navarro-Urrios et al. presents measurements of the complex nonlinear dynamics of a silicon optomechanical beam resonator. These dynamics arise from the interplay of optomechanical coupling, thermo-optic nonlinearity, and free-carrier dispersion. In particular, the authors observe and describe self-pulsing at integer fractions of the mechanical frequency and chaotic behaviour at high power. These are interesting observations, and the system highlights the rich physics that emerges in these systems coupling multiple nonlinearities with different dynamics. As such, I think they could be suitable for publication in Nature Communications.

I do have several questions and requests for clarification, that the authors would need to address in their manuscript to make it suitable for publication:

- After referring to other papers that describe chaos (references 11-14), the authors state "In contrast to the standard description, we report here on the non-linear dynamics of an OM cavity system (...) mainly affected by optical non-linearities of very diverse origin: thermo-optic effects, free-carrier dispersion, and OM coupling." This seems to be a statement of novelty, but the authors fail to mention that reference [14] claims precisely the same set of causes, and describes the same phenomena. The authors should more carefully and clearly describe how their work differs from reference [14], and what new insights are gained. At the very least, they should not give the impression that the aforementioned mechanisms are entirely novel and/or unique.

- The vertical dashed lines in Fig. 1b are not defined

- page 3, second column, line 1: the low phase noise of the RF signal is claimed; can this be quantified?

- page 3, last line: how can it be understood or expected that the frequency differences increase for lower M values?

- page 4, second paragraph ("lambda-bistable regions unfold..."): This paragraph is not clear: It is even unclear why it would be necessary to operate in two windows. In fact, it is also not clear what two windows are meant: as it is formulated now, it seems to refer to (1) the experimental bistable region and (2) the region in simulations.
- page 4, second column: are the period-doubling bifurcations reproduced in the simulations (and if not, why not)?
- figure 4: the chaotic regime at large wavelengths extends to the maximum probed wavelength (1541 nm): what is the expectation beyond this wavelength; i.e. how large is the spectral window over which chaos is maintained?
- page 7, line 8: the authors claim their LLE (a measure for chaos) cannot be compared to those in reference [14], because there "LLE values are much larger than the typical frequencies of the system". I don't see how this is a problem in fact. Moreover, if the authors can make that relative comparison of LLE and frequency in reference [14], why can't they relatively compare the LLE in [14] to their own work? In any case, given the close relation of both works, it would be good to make a serious attempt at describing the differences. Not only as it regards additional or different conclusions/insights, but also in terms of the magnitude of the effects or the parameters underlying them.

Response letter:

**Please take into account that reference numbering is associated to the original version of the manuscript. The font color of our responses is purple while actions taken are in orange.*

Reviewers' comments:

Reviewer #1 (Remarks to the Author):

The paper by Navarro-Urrios et al. reports measurements on an integrated one-dimensional optomechanical (beam) cavity composed of a clever designed Si-based photonic nanostructure with a lattice constant reduced towards the center of the beam. The authors analyze the dynamics of the system being mainly determined by optical-linearities due to thermo-optic, free-carrier dispersion and opto-mechanical coupling effects. Modifying exclusively the laser's power and wavelength they look for bi- and higher-dimensional limit cycles and, most interestingly, for a period doubling bifurcation route to chaos in optomechanical systems. Those states are precisely identified and characterized by RF spectra, see the wave-length forward and backward scans in Figs. 2 and 4 for moderate and high laser power, respectively. The interpretation of the results (and especially the onset of chaos) is flanked and supported by numerically simulations.

The time evolution of the transmitted signal below and above the bifurcation onsetting chaos (Fig. 5) is also interesting-looking: the periodic respectively aperiodic behavior is clearly demonstrated and the largest Lyapunov Exponent is extracted (in order to estimate for the amount of chaos).

In my opinion the study presented is a nice piece of experimental work on a really contemporary issue with potential applications in the rapidly developing field of photonics/phononics. The only point of criticism at this manuscript is that the authors have widely ignore previous theoretical work on non-linear dynamics and chaos in optomechanical systems. For example, the route to chaos in optomechanics considered in this paper has been worked out for paradigmatic optomechanical model system by Bakemeier et al in a recent PRL (114, 013601 (2015)) as well as optomechanical multistability and the existence of self-sustained oscillations (EPL 113, 64002 (2016)). The authors should definitely give credit to these papers and related work in the introduction and might relate their findings to these predictions (at least qualitatively) when discussing their data, before I can approve this paper for publication in Nature Communications.

*We agree with Reviewer #1 on the relevance of the suggested PRL reference, which is a theoretical work studying a route to chaos in an optomechanical (OM) system. **We have cited that reference in the new version of the manuscript.***

The route of chaos considered in the suggested PRL reference derives from the paradigmatic description of an optomechanical cavity, i.e., optical and mechanical coupled non-linear equations. Within this formalism, optomechanical "backaction" is the main ingredient for

activating high amplitude mechanical dynamics. Instead, the non-linear system of equations describing the OM system addressed in our work considers: thermo-optic (TO) effects, free-carrier dispersion (FCD) and OM coupling. It also assumes an adiabatic response of the optical cavity to temperature, free-carrier concentration or mechanical deformation. Thus, our system lacks optomechanical “backaction”, thus becoming essentially different from that considered in the suggested reference. Indeed, in our work we demonstrate both experimentally and numerically that it is also possible to predict an alternative route to chaos.

Regarding multistability, since our system is classical, *we have included ref. [F. Marquardt et al., Phys. Rev. Lett. 96, 103901, 2006]*, whose theoretical predictions were demonstrated recently in Ref. 6.

Reviewer #2 (Remarks to the Author):

The manuscript by Navarro-Urrios et al. presents measurements of the complex nonlinear dynamics of an silicon optomechanical beam resonator. These dynamics arise from the interplay of optomechanical coupling, thermo-optic nonlinearity, and free-carrier dispersion. In particular, the authors observe and describe self-pulsing at integer fractions of the mechanical frequency and chaotic behaviour at high power. These are interesting observations, and the system highlights the rich physics that emerges in these systems coupling multiple nonlinearities with different dynamics. As such, I think they could be suitable for publication in Nature Communications.

I do have several questions and requests for clarification, that the authors would need to address in their manuscript to make it suitable for publication:

- After referring to other papers that describe chaos (references 11-14), the authors state "In contrast to the standard description, we report here on the non-linear dynamics of an OM cavity system (...) mainly affected by optical non-linearities of very diverse origin: thermo-optic effects, free-carrier dispersion, and OM coupling." This seems to be a statement of novelty, but the authors fail to mention that reference [14] claims precisely the same set of causes, and describes the same phenomena. The authors should more carefully and clearly describe how their work differs from reference [14], and what new insights are gained. At the very least, they should not give the impression that the aforementioned mechanisms are entirely novel and/or unique.

We appreciate this comment made by Reviewer 2. He/she perceived that specific sentence as a statement of novelty, which was not our intention. For standard description we referred to the formalism used commonly to describe optomechanical cavities and we only meant to state that we use a much less common formalism. As the reviewer points out, the novelty of our work is not found in the nonlinear effects at stake, but in the rich set of stable solutions disclosed by our system, which are thoroughly reported both experimentally and by numerical modeling along the manuscript.

This misunderstanding has been amended by rephrasing the sentence in the manuscript in the following terms:

“We report here on the non-linear dynamics of an OM cavity system (...) mainly affected by optical non-linearities of very diverse origin: thermo-optic effects, free-carrier dispersion, and OM coupling.”

More details on the non-linear mechanisms and all issues concerning reference 14 are discussed later on a specific section at the end of the letter called “Comparison between our work and Ref. [14]”.

- The vertical dashed lines in Fig. 1b are not defined

Vertical dashed lines correspond to the spectral position of the simulated optical modes.

We have amended the manuscript accordingly by including the following text in the caption of the figure.

“Dashed lines indicate the spectral position of the simulated photonic modes.”

- page 3, second column, line 1: the low phase noise of the RF signal is claimed; can this be quantified?

Yes, the low phase noise of the tetra-dimensional limit cycle can be quantified in terms of the main RF peak linewidth, which in the specific case of the M=1 SP/phonon lasing state was about 2 kHz at $\lambda_l=1534\text{nm}$ and $P_l=2\text{mW}$, as extracted by setting a resolution bandwidth of 1kHz in our Spectrum Analyzer (see Figure R1). In the pure self-pulsing regime, the main RF peak linewidth is three orders of magnitude greater.

We have added the following text in the manuscript:

On page 3: “..., which displays a main RF peak linewidth of few kHz”.

Figure R1. Main RF peak (in linear scale) corresponding to a tetra-dimensional limit cycle state (black) and pure self-pulsing (red). The peaks have been normalized for comparison.

- page 3, last line: how can it be understood or expected that the frequency differences increase for lower M values?

This feature is both observed experimentally and by solving the numerical model. In order to give an intuition to it we refer to equations S4 and S5 of the Supplementary Information, which describe the dynamics of our non-linear system. The four differential equations are coupled through the number of intracavity photons (n), which depends on the relative detuning between the cavity resonance (λ_r) and the laser. Taking into account the dominant non-linear effects, i.e., Thermo-Optic (TO), Free-Carrier Dispersion (FCD) and Optomechanical coupling

(OM), we can express λ_r in the following terms:
$$\lambda_r \approx \lambda_o - \frac{\partial \lambda_r}{\partial N} N + \frac{\partial \lambda_r}{\partial T} \Delta T + \frac{\lambda_o^2 g_{o,OM}}{2\pi c} u_1$$

The back-coupling between equation S5 and equations S4 is the direct responsible for the comparison of frequency-entrained wide regions (frequency-plateaus) in the RF spectra. This back-coupling depends exclusively on the OM contribution, i.e., the last term of the previous equation. Thus, when the system is found in a SP/phonon lasing state, the OM contribution is of the same order of importance than the other terms while in a pure SP state it can be neglected. The previous equation also states that the OM coupling increases with the amplitude of the mechanical oscillation $u_{1,max}$. On the other hand, M represents the order of the harmonic of the optical force driving the mechanical motion, e.g., the M=1 phonon lasing state uses the main harmonic of the optical force. Since the intensity of those harmonics decrease with M (see for instance black curve of Fig. 4d), $u_{1,max}$ decreases as well. As a consequence, the OM coupling term decreases for higher M values, which leads to narrower frequency-entrained regions and smaller frequency differences.

We have added the following text in the manuscript:

In page 4, 1st paragraph, "in response to an increasing of the coupling strength between the $\{\Delta T, N\}$ system and the mechanical oscillator."

- page 4, second paragraph ("lambda-bistable regions unfold..."): This paragraph is not clear: It is even unclear why it would be necessary to operate in two windows. In fact, it is also not clear what two windows are meant: as it is formulated now, it seems to refer to (1) the experimental bistable region and (2) the region in simulations.

We are grateful to the reviewer for pointing out that this paragraph might be confusing in the terms stated in his/her comment. In addition, we are now convinced that it was probably not the best place of the manuscript to include that discussion. The point we wanted to make was that, in our system, it is possible to have equivalent dynamics in as many optical channels as photonic resonances are supported by the cavity, provided that only one resonance is pumped and the others are just probed without inducing further nonlinearities. Indeed, one of the main claims of Ref. [13] is that it was possible to transfer chaos from a strong signal to a very weak signal via mechanical motion, such that the signals are correlated and follow the same route to chaos.

To avoid confusion, we have removed that paragraph and added the following:

Page 3, 2nd paragraph: ", which is also observed in the numerical simulations (see Supplementary Discussion 6)."

Page 6, 2nd paragraph: "It is worth noting that, by weakly probing another photonic resonance in addition to the one being pumped, the chaotic dynamics of the pump signal will be transferred to the probe, since each one just follows the common dynamics of the effective refractive indices [13]."

- page 4, second column: are the period-doubling bifurcations reproduced in the simulations (and if not, why not)?

The period-doubling bifurcations are not reproduced in the simulations. Although our numerical model describes most of the experimental features observed, the referee correctly points out one that is not covered. The reason for that is that the current version of the model predicts that the frequency plateaus for $M > 1$ are much narrower than what observed, because the harmonics of the optical force are expected to be less effective than in the real world. Thus, as stated before, between two consecutive plateaus the simulations predict that the system passes through pure self-pulsing states, while, at high power, the experiments show that the width of the $M=2$ plateau is high enough to enable a transition between the $M=2$ and $M=1$ plateaus that does not involve pure SP states. Along the transition, the dimension of the system is, at least, four, following a route along which the attractor undergoes subsequent period-doubling bifurcations (see lower part of Fig. 4a, Fig. 4b and Fig. 4c).

This information was already included in Section S6 of the Supplementary Information (page 12) in the following terms:

"The model described by Eqs. S4 and S5 captures most of the features described along the main text. However, it does not incorporate three main experimental observations, namely: i) the period doubling bifurcation route between $M=2$ and $M=1$, ii) the presence of bistability and hysteresis between tetra-dimensional SP/phonon lasing and chaos and iii) the specific peak structure of the chaotic RF signal. The first observation remains a subject for further studies, since we predict much smaller high order plateaus than what measured experimentally, which eventually prevents reaching a similar situation to that reported in Fig. 4c. In the following section we address the former observations."

- figure 4: the chaotic regime at large wavelengths extends to the maximum probed wavelength (1541 nm): what is the expectation beyond this wavelength; i.e. how large is the spectral window over which chaos is maintained?

As it is shown in Fig. 4, the chaos dynamics is robust over a wide spectral range. The whole dynamics is wiped out slightly above 1541nm, when the second unstable fixed point showed in Fig. S8 gets close enough to the strange attractor volume, effectively deflecting down the trajectory of the system (like in Fig. S8d). The cavity then relaxes down to the initial "cold" situation at $\Delta T=0$, where the cavity resonance is far away from the laser line. In the measurements shown in Fig. 4 we have not reached that condition, since by doing so we would have lost the possibility of performing the back-scan measurement of Figure 4b.

In order to address this point, we have added the following text in the Supplementary Information, Section 6:

"The chaotic dynamics is robust along a wide spectral range, and is only wiped out when the second unstable fixed point showed in Fig. S8 gets close enough to the strange attractor volume, effectively deflecting down the trajectory of the system (like in Fig. S8d). The cavity then relaxes down to the initial "cold" situation at $\Delta T=0$, where the cavity resonance is far

away from the laser line. In the measurements shown in Fig. 4 we have not reached that condition, since by doing so we would have lost the possibility of performing the back-scan measurement of Figure 4b."

- page 7, line 8: the authors claim their LLE (a measure for chaos) cannot be compared to those in reference [14], because there "LLE values are much larger than the typical frequencies of the system". I don't see how this is a problem in fact. Moreover, if the authors can make that relative comparison of LLE and frequency in reference [14], why can't they relatively compare the LLE in [14] to their own work? In any case, given the close relation of both works, it would be good to make a serious attempt at describing the differences. Not only as it regards additional or different conclusions/insights, but also in terms of the magnitude of the effects or the parameters underlying them.

- Comparison between our work and Ref. [14]:

Reference 14 is a quite recent Arxiv paper (from August 2016), which is still not published in any journal to the best of our knowledge. We uploaded a version of our manuscript in the Arxiv platform just few weeks afterwards. We included it as a reference since it is one of the few experimental works claiming the observation of chaos in an optomechanical integrated system.

In our opinion, reference 14 contains several controversial aspects, which will be discussed in the following paragraphs. It consists of a main text that addresses several times to a Supplementary Information document for eventual details. However, the Supplementary Information file has not been uploaded to Arxiv. Therefore, there is not enough information to discuss some of the arguable statements of that reference in depth and compare them with our findings. In any case, hereafter we enlist several points that may allow comparing the two manuscripts better:

- The studied geometries are different. Ref 14 studies on a 2D crystal by exploiting the OM interaction between a slot optical mode and a single breathing mechanical mode. On the other hand, we report on a 1D nanobeam in which several mechanical modes are at stake, enriching some of the dynamical solutions. Indeed, the chaotic states reported in Figs. 4 and 5 involve the activation of an in-plane and an out-of-plane mechanical modes, thus making the chaotic state intrinsically different from that claimed in Ref. 14. Two in-plane flexural modes are also involved in the laser power bi-stability reported on Fig. 3.
- One of the main points of our paper is the observation of bistability and hysteresis between bidimensional and tetra-dimensional limit-cycles, between different coherent mechanical states and between tetra-dimensional limit-cycles and chaos. Both the laser wavelength and its power have been varied in different senses to unveil those features. We have also successfully reproduced those features by solving a numerical model of non-linear differential equations. Ref. 14 does not address bi-stability of any kind.

- Referee 2 is right when he/she states that the basic underlying mechanism leading to the dynamics observed in Ref. 14 is Thermo-Optic/Free-Carrier-Dispersion (TO/FCD) self-pulsing (SP) driven by two-photon absorption. However, as we acknowledge in our manuscript, this dynamics is not novel. Pure self-pulsing has been reported in several exclusively-photonics systems during the last decade (see pages 4-6 of Ref. 18 for a recent review on the topic). To the best of our knowledge, it was first observed a decade ago by Johnson et al. (ref. 19, Opt. Express, 14, 817-831 (2006)). In Ref. 7 (Navarro-Urrios et al. Sci. Reports 5, 2015) we experimentally demonstrate for the first time that SP can couple to a mechanical mode through optical forces in an optomechanical system. As a consequence of that coupling, we also demonstrate “mechanical lasing”. Those experimental observations were reproduced with a numerical model, reported for the first time in our Ref. 7 in 2015, which is essentially equivalent to that of Ref. 14 (Eqs. 1-4). Reference 14 ignores any reference to SP, including our work from Ref. 7. It is also worth noting the same group have published a conference paper earlier this year concerning an equivalent device (see Y. Huang, J. G. Flores, Z. Cai, M. Yu, D. Kwong, G. Wen, and C. W. Wong, “Controllable optomechanical coupling and Drude self-pulsation plasma locking in chip-scale optomechanical cavities,” in Conference on Lasers and Electro-Optics, OSA Technical Digest (online) (Optical Society of America, 2016), paper FTu3B.8.), where they acknowledge our Ref. 7 in the following terms: “...when driving the cavity into oscillation state (the corresponding optical transmission can be found in the inset of Fig. 2(d)), both subharmonics (1/6) and harmonics can be found due to the strong interaction between the OMO and SP [5.]”, [5] being a citation to our Ref. 7.
- Regarding the interpretation of the experimental observations, the several features of Figure 1e of Ref. 7 are just barely described. First they report “low-threshold optomechanical motion”, which is widely acknowledged as the optical transduction of motion driven by thermal Langevin forces. By increasing the laser wavelength, the system displays unstable pulses (USP, the authors do not provide further insights) and then the system passes through several flat frequency regions at integer fractions of the mechanical modes. The interpretation of the dynamics within those regions is also missing in Ref. 14. We have already reported similar flat frequency regions in Ref. 7 together with the result of a numerical modelling that reproduces the experimental findings and the following interpretation: “When the mechanics/self-pulsing resonant condition is achieved, “flat regions” appear, indicating the coherent vibration of the OM photonic crystal. Since all the dynamics is coupled together, the OM oscillations provide an active feedback that stabilizes the SP. In those specific conditions, the two oscillators are frequency-entrained (FE) in a way that the SP adapts its oscillating frequency to be a simple fraction of the mechanical eigenfrequency. Similarly to the case of synchronized oscillators, the lowest M values have the largest FE zones....”.
- Regarding the observation of chaos, which is a feature that both papers claim, at specific laser-cavity detunings, the cavity of Ref. 14 enters into a chaotic state. However, the authors of Ref. 14 misinterpret the transition among all the previous states as a route towards chaos, which is completely misleading. A typical route to chaos is studied, for instance, in Ref. [L. Bakemeier, A. Alvermann, and H. Fehske, Phys. Rev. Lett. 114, 013601, 2015], where the dynamical system starts from a limit cycle and becomes chaotic through subsequent period doubling bifurcations as an external parameter is varied. Along the route, the effective dimension of the system must be always higher than two, as stated by the Poincaré-Bendixson

theorem. Contrary to what is stated by the authors of Ref. 14, a SOM state (isolated SP) cannot participate in this route because of being a solution of an effective bidimensional system. Indeed, in our opinion, the route to chaos in the case of Ref. 14 should have been studied just in the transition between the $f_{\text{OMO}}/2$ state and the chaos state. Therefore, we believe that Figures 3 and 4e are completely misleading as what reported is not a route to chaos.

In our case, we clearly state that the transition to chaos occurs only at high enough powers, is abrupt to the best of the resolution of our tunable laser (1 pm) and starts from the $M=1$ coherent state. We also observe that the transition between the $M=2$ and $M=1$ at high power follows a route along which the attractor undergoes subsequent period-doubling bifurcations (see lower part of Fig. 4a, Fig. 4b and Fig. 4c).

- Finally, regarding the analysis of the temporal signals of Ref. 14, it is stated that the details would be provided in the missing Supplementary Information document, which prevents us to do a complete analysis. Authors of Ref. 14 use the Grassberger-Procaccia (GPA) method, which is known to have been the most popular method used to quantify chaos in the 80's. However, as stated in a later reference (Ref. [20]), it is sensitive to variations in its parameters, e.g., number of data points, embedding dimension, reconstruction delay, and it is usually unreliable except for long, noise-free time series. Hence, the outcomes of the GPA algorithm are questionable. The authors of Ref. [14] do not give details of the time registers in the main text, i.e., number of points and total temporal length evaluated. Moreover, the largest Lyapunov exponent (LLE) provided by the authors of Ref. 14 is 3 times greater than the typical frequency of the system, which means that the horizon of predictability is much smaller than the typical period of the system. Just by inspection of the temporal series of Figure 2a of Figure 3d2 it is possible to realize that this cannot be the case of Ref. [14], i.e., the signal should look almost stochastic and it is not. Therefore, we think that the quantification of chaos described in Ref. [14] is, at least, controversial.

In our case, we apply the Rosenstein algorithm, which is widely acknowledge to be accurate because it takes advantage of all the available data and works well with small data sets. In addition, it is robust where GPA is not, i.e., to changes in the embedding dimension, size of data set, reconstruction delay, and noise level.

To summarize the previous discussion, we find that Ref. 14 contains many controversial and incomplete results, which have not been positively refereed yet. On the other hand, we also believe that we have demonstrated that our work clearly distinguishes from Ref. 14. We believe that including a discussion of the similarities or differences of our system with ref 14 will not contribute positively to the current manuscript and will delude its main focus and conclusions. Thus we have not made further amendments and we sincerely hope that Referee 2 will agree with our decision.

Reviewers' comments:

Reviewer #1 (Remarks to the Author):

In my opinion, the revised manuscript adequately accommodates the comments/suggestions addressed by the referees; I recommend its publication in Nature Communications.

Reviewer #2 (Remarks to the Author):

I have carefully reviewed the answers of the authors to my earlier comments. I think they have satisfactorily responded to all suggestions, with the exception of the last point. There, I essentially asked two things:

(1) if the authors could make a serious attempt at describing the differences with reference [14], given its close relationship to their work.

(2) if they could clarify how it can be justified that they cannot compare their LLE to the LLE in reference [14], even though they do make a quantitative statement about the LLE in reference [14] (by saying it is larger than the frequency).

The authors respond with a lengthy statement describing why they take issue with several of the analyses performed in reference [14], and why their work differs from it. But they refuse to make any attempt at clarifying that difference in the manuscript. I think that is a pity. Even if they believe some of the analysed conclusions cannot be compared, a comparison of the observed phenomena (the different regimes and their orders) as well as the parameters at which these are obtained (optical powers, intrinsic system parameter) would have been possible, as it would only rely on reported measurements. It would have been insightful to readers who would like to study phenomena such as these in comparable (or other) systems.

Regarding question (2), I now understand the following from the response: The authors in fact do not believe that the LLE reported in reference [14] is correct (as it is retrieved through a different, inferior, method). But they do not want to say this in their paper. Instead, they now write in essence that the LLEs cannot be compared because the LLE of reference [14] is so large. But such an argument is impossible to follow and self-contradictory. As such, the paragraph cannot be left in as it is. A potential solution could for example be: To explain that the LLE in reference [14] was obtained using a different method, that the authors chose to use the Rosenstein algorithm because of its reliability, that it would be interesting to compare different systems on equal footing in the future, and that it would be good to generally determine what kind of optomechanical systems give rise to the strongest chaos at lowest power.

Response letter to Reviewer #2:

Reviewer #2 (Remarks to the Author):

I have carefully reviewed the answers of the authors to my earlier comments. I think they have satisfactorily responded to all suggestions, with the exception of the last point. There, I essentially asked two things:

- (1) if the authors could make a serious attempt at describing the differences with reference [14], given its close relationship to their work.
- (2) if they could clarify how it can be justified that they cannot compare their LLE to the LLE in reference [14], even though they do make a quantitative statement about the LLE in reference [14] (by saying it is larger than the frequency).

The authors respond with a lengthy statement describing why they take issue with several of the analyses performed in reference [14], and why their work differs from it. But they refuse to make any attempt at clarifying that difference in the manuscript. I think that is a pity. Even if they believe some of the analysed conclusions cannot be compared, a comparison of the observed phenomena (the different regimes and their orders) as well as the parameters at which these are obtained (optical powers, intrinsic system parameter) would have been possible, as it would only rely on reported measurements. It would have been insightful to readers who would like to study phenomena such as these in comparable (or other) systems.

Regarding question (2), I now understand the following from the response: The authors in fact do not believe that the LLE reported in reference [14] is correct (as it is retrieved through a different, inferior, method). But they do not want to say this in their paper. Instead, they now write in essence that the LLEs cannot be compared because the LLE of reference [14] is so large. But such an argument is impossible to follow and self-contradictory. As such, the paragraph cannot be left in as it is. A potential solution could for example be: To explain that the LLE in reference [14] was obtained using a different method, that the authors chose to use the Rosenstein algorithm because of its reliability, that it would be interesting to compare different systems on equal footing in the future, and that it would be good to generally determine what kind of optomechanical systems give rise to the strongest chaos at lowest power.

We appreciate the comments and suggestions made by Reviewer 2, which we understand are meant to improve the quality and impact of the manuscript.

Question (1)

In order to address this issue, which is shared by the editor, we have included in the main text of the article a direct comparison between our system and that of reference 14, including a quantitative comparison of the optical energy stored in the cavity for disclosing the chaotic regime. In addition to that, we have included a comprehensive comparison between both works in the Supplementary Material. The added section is largely based on the statements provided in our previous response letter. This discussion now includes quantitative comparisons where it is possible,

The following text is added in the main manuscript.

Page 4, 2nd column, 2nd paragraph: "... , i.e., an intracavity stored optical energy (calculated at perfect laser/cavity resonance) of 50 fJ, which is slightly lower than the energy required to disclose a chaotic regime in Ref. [16] (60fJ). In our case,..."

Page 7, 2nd column, 2nd paragraph: "It is worth noting that a recent reference [Ref. 16] reports on a 2D integrated geometry showing similarities to our system in terms of some of the observed dynamical solutions. In addition to the claim of chaotic dynamics, which has been discussed in comparison to our findings in the previous section, [Ref. 16] reports pure self-pulsing and several "phonon lasing" regimes, each one activated using different harmonics of the optical force. Those latter features are in agreement with what reported by us previously in Ref. [18]. A comprehensive comparison of the works of both groups, including an analysis of the input parameters at which the previous regimes are triggered is included in the Supplementary Discussion 9."

Hereafter we transcript the new section (Section 9) added to the Supplementary Material file.

Comparison between our work and Ref. [J. Wu et al., arXiv:1608.05071 (2016); Ref. [16] of the main text and Ref. (10) of the Supplementary Information file]*:

Reference (10) is one of the few experimental works claiming the observation of chaos in an optomechanical integrated system. Hereafter we enlist several points that may allow comparing it with our works, including the mechanisms reported in the current manuscript and previous reports, i.e. ref. (7):

- **Comparison of geometries.** Ref. (10) studies a 2D crystal by exploiting the OM interaction between a slot optical mode and a single breathing mechanical mode. On the other hand, we report on a 1D nanobeam in which several mechanical modes are at stake, enriching some of the dynamical solutions. Indeed, the chaotic states reported in Figs. 4 and 5 involve the activation of an in-plane and an out-of-plane mechanical modes, thus making the chaotic state intrinsically different from that claimed in Ref. (10). Two in-plane flexural modes are also involved in the laser power bi-stability reported on Fig. 3 of the main text.

- **Bistability and hysteresis.** One of the main points of our manuscript is the observation of bistability and hysteresis between bidimensional and tetra-dimensional limit-cycles, between different coherent mechanical states and between tetra-dimensional limit-cycles and chaos. Both the laser wavelength and its power have been varied in different senses to unveil those features. We have also successfully reproduced those features by solving a numerical model of non-linear differential equations. Ref. (10) does not address bi-stability of any kind.

- **Comparison of basic dynamics: Pure self-pulsing and “phonon lasing”.** The basic underlying mechanism leading to the dynamics observed in Ref. (10) is Thermo-Optic/Free-Carrier-Dispersion (TO/FCD) self-pulsing (SP) driven by two-photon absorption. However, as we acknowledge in our manuscript, this dynamics is not novel. Pure SP has been reported in several exclusively-photonic systems during the last decade (see pages 4-6 of Ref. (11) for a recent review on the topic). To the best of our knowledge, it was first observed a decade ago by Johnson et al. (12). In Ref. (7) we experimentally demonstrate for the first time that SP can couple to a mechanical mode through optical forces in an optomechanical system. As a consequence of that coupling, we also demonstrate “phonon lasing”. Those experimental observations were reproduced with a numerical model, reported for the first time in our Ref. (7) in 2015, which is essentially equivalent to that of Ref. (10) (Eqs. 1-4). Reference (10) omits any reference to SP, including our work from Ref. (7).

The maximum frequency reached by the pure SP regime is similar in both systems (several tens of MHz) as we understand it is mainly limited by heat dissipation to the surrounding atmosphere.

- **Comparison of the interpretation of the different dynamic regimes.** By increasing the laser wavelength, the system of Ref. (10) displays unstable pulses (USP, the authors do not provide further insights), which are not present in our system. Then the system of Ref. (10) passes through several flat frequency regions at integers fractions of the mechanical modes. The interpretation of the dynamics within those regions is missing in Ref. (10). In 2015, we reported for the first time similar, though much wider, flat frequency regions in Ref. (7) at comparable laser powers together with the result of a numerical modelling that reproduces the experimental findings and the following interpretation: “*When the mechanics/self-pulsing*

resonant condition is achieved, “flat regions” appear, indicating the coherent vibration of the OM photonic crystal. Since all the dynamics is coupled together, the OM oscillations provide an active feedback that stabilizes the SP. In those specific conditions, the two oscillators are frequency-entrained (FE) in a way that the SP adapts its oscillating frequency to be a simple fraction of the mechanical eigenfrequency. Similarly to the case of synchronized oscillators, the lowest M values have the largest FE zones....”.

- **Comparison of OM interaction strengths.** The only mechanical mode at play in Ref. (7) happens at almost twice the maximum SP frequency (i.e. at 112 MHz) and its OM coupling rate is stated to be $g_{o,OM}/2\pi = 110$ kHz. In our case, the main mechanical mode that is at the heart of the complex dynamics discussed along the manuscript, is an in-plane flexural mode happening at 54 MHz, which falls within the frequency range covered by the pure SP dynamics, and displaying an OM coupling rate of $g_{o,OM}/2\pi = 300$ kHz. This two combined values enable a much stronger coupling between the SP and the mechanical harmonic oscillator. Indeed, in our work it is possible to use the first harmonic of the optical force to drive the mechanical mode. The stronger coupling appears evident when comparing the much larger frequency entrained regions (frequency plateaus) present in our case (4-6 nm in the best case) with those reported in Ref (10) (less than 1 nm in the best case).
- **Qualitative comparison of the chaotic dynamics and route towards chaos.** Ref. (10) claims the observation of chaos at specific laser-cavity detunings. In our opinion, the authors of Ref. (10) misinterpret the transition among all the previous states as a route towards chaos. A typical route to chaos is studied, for instance, in Ref. (13), where the dynamical system starts from a limit cycle and becomes chaotic through subsequent period doubling bifurcations as an external parameter is varied. Along the route, the effective dimension of the system must be always higher than two, as stated by the Poincarè-Bendixson theorem. Contrary to what is stated by the authors of Ref. (10), a SOM state (isolated SP) cannot participate in this route because of being a solution of an effective bidimensional system. Indeed, in our opinion, the route to chaos in the case of Ref. (10) should have been studied just in the transition between the $f_{OMO}/2$ state and the chaos state. Therefore, our opinion is that Figures 3 and 4e of Ref. (10) do not report a route to chaos.
In our case, we clearly state that the transition to chaos occurs only at high enough powers, is abrupt to the best of the resolution of our tunable laser (1 pm) and starts from the $M=1$ coherent state. We also observe that the transition between the $M=2$ and $M=1$ at high power follows a route along which the attractor undergoes subsequent period-doubling bifurcations (see lower part of Fig. 4a, Fig. 4b and Fig. 4c).
- **Comparison of intracavity optical energy thresholds to disclose chaotic dynamics.** An intracavity stored optical energy (calculated at perfect laser/cavity resonance) of about 50 fJ is required for observing chaos, which is slightly lower than in Ref. (10) (60fJ).
- **Quantitative comparison of the analysis of the chaotic dynamics.** Authors of Ref. (10) use the Grassberger-Procaccia (GPA) method, which is known to have been the most popular method used to quantify chaos in the 80's. However, as stated in a later reference (Ref. [22]), it is sensitive to variations in its parameters, e.g., number of data points, embedding dimension, reconstruction delay, and it is usually unreliable except for long, noise-free time series. Hence, the outcomes of the GPA algorithm are questionable. The authors of Ref. (10) do not give details of the time registers in the main text, i.e., number of points and total temporal length

evaluated. Further details would be provided in the missing Supplementary Information document, which prevents us to do a complete analysis. Moreover, the largest Lyapunov exponent (LLE) provided by the authors of Ref. (10) is 3 times greater than the typical frequency of the system, which means that the horizon of predictability is much smaller than the typical period of the system. Just by inspection of the temporal series of Figure 2a of Figure 3d2 it is possible to realize that this cannot be the case of Ref. (10), i.e., the signal should look almost stochastic and it is not.

In our case, we apply the Rosenstein algorithm, which is widely acknowledge to be accurate because it takes advantage of all the available data and works well with small data sets. In addition, it is robust where GPA is not, i.e., to changes in the embedding dimension, size of data set, reconstruction delay, and noise level.

****By the time of submitting the current manuscript, the Supplementary Information file of Ref. (10) has not been made public.***

Question (2)

We agree with Reviewer 2 that our statement in what concerns the comparison with LLE values estimated in Ref. [14] has to be revised and extended.

The following paragraph placed in page 7:

“Notice that it was not possible for us to compare the obtained LLE values to those reported elsewhere for OM integrated systems. In Ref. [15], those are reported without units and in Ref. [16] LLE values are much larger than the typical frequencies of the system, i.e., the horizon of predictability is much shorter than the time required to perform a single oscillation.”

has been replaced by the following:

“An estimation of the LLE in OM systems has been realized in Refs. [15] and [16]. In the former reference, its value is provided without units, preventing any comparison with our work. In the case of Ref. [16], the LLE exceeds by three orders of magnitude the one reported here and is much larger than the typical frequencies of the system. In that work, the LLE is calculated using the Grassberger-Procaccia (GPA) method proposed in Ref. [24], which, as stated in a later reference (Ref. [20]), is sensitive to variations of its input parameters (number of data points, embedding dimension, reconstruction delay) and it is usually unreliable except for long, noise-free time series. In our case, we apply the ROS algorithm, which is robust where GPA is not and is widely acknowledge to be accurate because it takes advantage of all the available data and works well with small data sets.”

We have also added a new reference to support the previous discussion: *Grassberger, P., Procaccia, I. Characterization of strange attractors. Phys. Rev. Lett. 50, 346 (1983).*